# Identifying Homogeneous and Interpretable Groups for Conformal Prediction

**Natalia Martinez Gil**[1]    **Dhaval Patel**[1]    **Chandra Reddy**[1]    **Giridhar Ganapavarapu**[1]    **Roman Vaculin**[1]

**Jayant Kalagnanam**[1]

[1]IBM Research, Yorktown Heights, New York, USA

## Abstract

Conformal prediction methods are a tool for uncertainty quantification of a model's prediction, providing a model-agnostic and distribution-free statistical wrapper that generates prediction intervals/sets for a given model with finite sample generalization guarantees. However, these guarantees hold only on average, or conditioned on the output values of the predictor or on a set of predefined groups, which a-priori may not relate to the prediction task at hand. We propose a method to learn a generalizable partition function of the input space (or representation mapping) into interpretable groups of varying sizes where the nonconformity scores - a measure of discrepancy between prediction and target - are as homogeneous as possible when conditioned to the group. The learned partition can be integrated with any of the group conditional conformal approaches to produce conformal sets with group conditional guarantees on the discovered regions. Since these learned groups are expressed as strictly a function of the input, they can be used for downstream tasks such as data collection or model selection. We show the effectiveness of our method in reducing worst case group coverage outcomes in a variety of datasets.

## 1 INTRODUCTION

The interest on the application of Machine Learning (ML) models on different industrial settings has increased in recent years, in particular given the success of deep neural networks and the availability of large amounts of data. In general, predictive ML models are optimized to capture the behaviour of a target variable based on a finite set of observations. One of the major concerns when deploying these models into real-world decision making processes is how to quantify the uncertainty in their prediction, especially in high-stakes domains such as health care or finance where there is a robust penalty for making mistakes.

Typical ML models produce point predictions (e.g., expected values for regression or most likely label in the case of classification), these are not a-priori informative on the range of values the target variable can take within normal operation (i.e. set of values expected to occur with high probability). Calibrated prediction sets (or interval) can be of great value for a decision maker that wants to consider worst-case scenarios. Moreover, understanding how the uncertainty of a model's prediction differs across varying subsets of the available data can inform the data collection process, model improvements or model selection/assessment.

Conformal prediction methods Vovk et al. [2005] have gained significant popularity in recent years since they offer a distribution-free approach to quantify the uncertainty of a black box model's prediction with generalization guarantees Shafer and Vovk [2008], Angelopoulos and Bates [2021]. In particular, split conformal prediction (SCP) Papadopoulos et al. [2002] is an attractive post-hoc, model-agnostic approach that only requires access to the model's prediction and a calibration dataset. This is especially useful in settings where retraining or modifying an ML model to produce uncertainty estimates is infeasible, or when only query access to an ML model is possible (e.g., LLMs,[1]).

Given a desired miscoverage level $\alpha$ (i.e. error-rate) conformal prediction methods produce prediction sets/intervals based on a black box model's prediction that are guaranteed to contain, on average, the ground truth value of the target variable with probability larger or equal than $1 - \alpha$. They often rely on the quantile estimation of a non-conformity score, which is a measure of the disagreement between the target variable and the model prediction (e.g., absolute error), and only require that the calibration dataset be exchangeable[2] with the data samples the model will be tested on. Different

---

[1]Large Language Models

[2]This is a weaker condition than full statistical independence

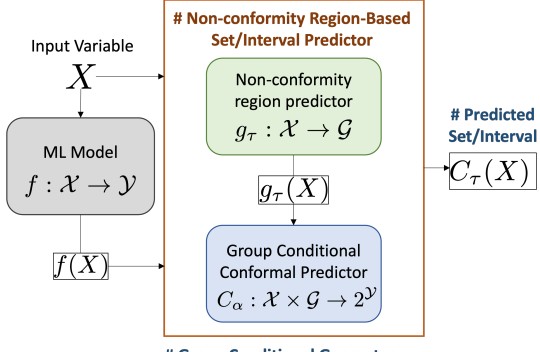

(a) Non-conformity region-based prediction framework

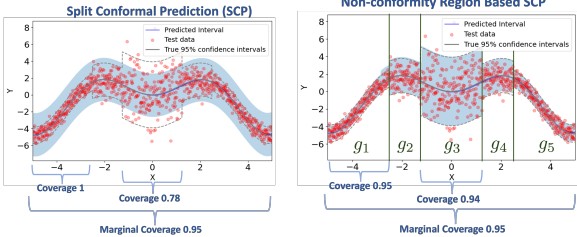

(b) Split Conformal Prediction      (c) Region-based SCP

Figure 1: (1a) Overview of proposed framework to produce prediction intervals/sets. We first decompose the input space into interpretable groups where each group contains homogeneous predictions of the $1 - \alpha$-th quantile of the non-conformity scores of a ML model's prediction $f(\cdot)$. We then build a prediction interval/set, denoted $C_\tau(X) = C_\alpha(X, g_\tau(X))$ where $C_\alpha$ is the group-conditional conformal predictor, which depends on both the input $X$ and on the group prediction $g_\tau(X)$. $C_\tau$ satisfies group conditional coverage guarantees for the identified groups. (1b) regression example of heteroskedastic uncertainty in the model's prediction (blue line), the x-axis indicates the input variable, y-axis the target variable, and red dots the test samples. Prediction bands (blue) are produced by standard SCP with a coverage target of 0.95 ($\alpha = 0.05$), the desired coverage is achieved on average but there is significant disparity across regions of $X$. (1c) shows the prediction bands obtained with the proposed region-based approach in conjunction with SCP. Five groups where identified, and the group conditional coverage is improved significantly w.r.t. SCP.

works have studied how to adapt these methods to scenarios where the exchangeability assumption is violated, such as distribution shifts or time series settings Gibbs and Candes [2021], Stankeviciute et al. [2021], Barber et al. [2023].

A significant amount of work has focused on understanding the feasibility of more efficient prediction sets and stronger-than-average coverage guarantees. An ideal goal would be to achieve input-conditional coverage (i.e., the coverage guarantees hold for each possible input), which has been proven to be impossible in practice Vovk [2012], Lei and Wasserman [2014]. Nonetheless, weaker guarantees such as local conditional coverage Foygel Barber et al. [2021] or group and level-set conditional coverage Jung et al. [2022] are possible. Providing predictions sets with close to conditional coverage guarantees is valuable in settings where the model's prediction uncertainty differs significantly across the input space (heteroskedastic uncertainty). Essentially, we want to avoid having subsets of samples with under coverage and/or inefficient prediction sets Romano et al. [2020]. Marginal-coverage guarantees hold only on average, and do not prevent high variation in the performance of the prediction sets across subgroups in the input space.

Many works have addressed relaxations of the conditional coverage objective by modifying the non-conformity score Papadopoulos et al. [2011], Lei and Wasserman [2014], Guan [2023], Han et al. [2022], Amoukou and Brunel [2023], Seedat et al. [2023], Ghosh et al. [2023], learning the non-conformity quantile threshold Jung et al. [2022], Bastani et al. [2022], Gibbs et al. [2023], or using a conformal quantile regression objective when the provided model can be retrained Romano et al. [2019]. In particular, a line of work with practical guarantees has focused on the notion of local or group-conditional coverage for a pre-specified set of groups that partitions the input space Vovk et al. [2003], Vovk [2012] and for overlapping groups Foygel Barber et al. [2021], Jung et al. [2022], Gibbs et al. [2023].

**Main Contributions.** Most group conditional conformal prediction approaches presented above rely on pre-defined groups or propose greedy approaches to slice the input space Lei and Wasserman [2014] or the prediction space Sesia and Romano [2021], Boström et al. [2021] into equal-sized regions, which scale poorly to higher dimension inputs. To address this issue, we propose a method to learn a generalizable partition function of the input space (or representation mapping) into interpretable groups[3] of varying sizes where the quantiles of the non-conformity scores are as homogeneous as possible when conditioned to the group. The main characteristics of the proposed approach are described next.

- We adopt an adversarial approach where an agent proposes a partition function that approximates the non-conformity-score conditional quantile; and a judge then evaluates it based on its worst group conditional miscoverage with respect to the one achieved by an interpretable baseline. The agent and the judge use independent datasets drawn from the same distribution.

- We define a fitness score denoted as worst group miscoverage ratio (MCR) that allows the comparison of

---

[3]we use the terms regions, groups, partitions and clusters interchangeably

models across different partitions. We use this score to inform the regularization of a family of interpretable clustering functions with the goal of selecting the partition that best generalizes in terms of MCR over the set of partitions that accurately approximate the conditional quantile estimates of the non-conformity scores.

- We learn partitions using decision trees since the identified groups can be described based on interpretable input rules—a valuable property for downstream tasks such as data collection or model selection. The partition function can be integrated with any of the group conditional conformal approaches discussed previously (see Figure 1) to produce conformal sets with group conditional guarantees on the discovered regions.

The proposed method serves as an inexpensive alternative to a more strict and costly auditing approach where the auditor leverages an optimization procedure to find the worst computationally identifiable miscoverage group for a given model. In our experiments, we show that we discover meaningful groups that significantly benefit from their inclusion in a group conditional conformal approach. Code is available at `https://github.com/trustyai-explainability/trustyai-model-trust`.

**Manuscript Organization.**    Section 2 provides a summary of conformal prediction definitions that are used throughout this manuscript and Section 3 summarizes additional related work. Section 4 describes the proposed objective for discovering the group partition function based on non-conformity score quantiles. Section 5 provides the method that integrates group identification with conformal prediction. Finally, Section 6 shows experimental results that validate our proposed approach.

## 2   BACKGROUND

Let us consider the supervised machine learning setting where we have an input variable $X \in \mathcal{X}$ and a target variable $Y \in \mathcal{Y}$ jointly distributed according to an unknown distribution $X, Y \sim p(X)p(Y|X)$. Given a prediction function $f : \mathcal{X} \to \mathcal{Y}'$, where $\mathcal{Y}'$ is an output space that approximates some statistic of $p(Y|X)$, [4], we consider a non-conformity score function $S_f : \mathcal{X} \times \mathcal{Y} \to \mathbb{R}$, that depends on $f$ and measures the proximity between the prediction $f(X)$ and the corresponding target $Y$. We use $S$ to denote the non-conformity random variable $S = S_f(X, Y)$ that depends on the input variable $X$, target variable $Y$ and model $f$.

In the split conformal setting we assume we have access to an i.i.d. calibration dataset $\mathcal{D}^{cal} = \{(X_i, Y_i)\}_{i=1}^n \sim p(X, Y)^{\otimes n}$ that is independent of $f$. Then, the set of

---

[4]e.g., $\mathcal{Y}' = \Delta^{|\mathcal{Y}|-1}$ if $f$ outputs a probability vector over labels in the classification setting, or $\mathcal{Y}' = \mathcal{Y}$ for regression

non-conformity scores $\{s_i = S_f(X_i, Y_i)\}_{i=1}^n$ are exchangeable with any unseen non-conformity score sample $S_{n+1} = S_f(X_{n+1}, Y_{n+1})$. This exchangeabilty property implies the following for any new sample $X_{n+1}, Y_{n+1} \sim P(X)P(Y|X)$

$$1 - \alpha \le P\Big(S_{n+1} \le Q_{1-\alpha}(\{s_i\}_{i=1}^n)\Big) \le 1 - \alpha + \tfrac{1}{n+1}$$
$$Q_{1-\alpha}(\{s_i\}_{i=1}^n) = \mathbb{Q}_{1-\alpha}(\sum_{i=1}^n \tfrac{1}{n+1}\delta_{s_i} + \tfrac{1}{n+1}\delta_\infty) \tag{1}$$

where $\mathbb{Q}_{1-\alpha}(\cdot)$ denotes the $1-\alpha$ quantile operator of its input (for Eq.1 this is the $\lceil (1-\alpha)(n+1) \rceil$-th smallest $\{s_i\}_{i=1}^n$), and $\alpha \in (0, 1)$ is a pre-specified mis-coverage level. Conversely, we can define the conformity set for a given mis-coverage level $\alpha$ based on the non-conformity score function as

$$C_f(X_{n+1}) = \{Y_{n+1} \in \mathcal{Y} : S_{n+1} \le Q_{1-\alpha}(\{s_i\}_{i=1}^n)\}. \tag{2}$$

This satisfies $P(Y_{n+1} \in C_f(X_{n+1})) \ge 1 - \alpha$.

**Conditional and Local Coverage Guarantees**    The set described in Eq. 2 provides guarantees on average across the entire data distribution, but not for any particular value of $X$, i.e., $P(Y_{n+1} \in C_f(X_{n+1})|X_{n+1} = x) \ge 1 - \alpha, \forall x \in \mathcal{X}$, also known as conditional coverage. This desirable guarantee cannot be achieved in practice Vovk [2012], Lei and Wasserman [2014], Foygel Barber et al. [2021], since it would require $C_f(x)$ to have infinite expected length at any non-atom $x$. A relaxation of this setting is to consider local coverage over a partition of the support of $P(X)$ denoted as $g : \mathcal{X} \to \mathcal{G}$ with $\mathcal{G}$ a discrete finite set. Then, local conditional guarantees implies $P(Y_{n+1} \in C_f(X_{n+1})|X_{n+1} \in g_j) \ge 1 - \alpha$ with $g_j = \{x : g(x) = j\} \forall j \in \mathcal{G}$.

**Pinball Loss in the Infinite Sample Regime**    In the ideal case were the conditional distribution of the non-conformity scores $(P(S|X))$ is known, the most efficient prediction interval for a given $X$ and mis-coverage level $\alpha$ is

$$C(X) = \{y \in \mathcal{Y} : S(X, y) \le F_{S|X}^{-1}(1-\alpha)\} \tag{3}$$

with $F_{S|X}^{-1}(1-\alpha) = \inf\{\hat{s} \in supp(P_{S|X}) : P(S \le \hat{S}|X) \ge 1 - \alpha\}$. We can approximate the $1-\alpha$ conditional quantile by minimizing the expected pinball loss

$$F_{S|X}^{-1}(1-\alpha) = \arg\min_{q \in \mathcal{Q}} \mathbb{E}_{p(X,S)}\big[\ell_{1-\alpha}(q(X), S)\big] \tag{4}$$

where $\mathcal{Q}$ represents a universal class of functions and $\ell_{1-\alpha}(\cdot, \cdot)$ is the pinball loss, defined as

$$\ell_{1-\alpha}(q, s) = \max\{(1-\alpha)(s-q), \alpha(q-s)\}. \tag{5}$$

Section 4 leverages the pinball loss, in addition to a worst case generalization objective, to identify a set of disjoint regions in the input space where the $1-\alpha$ quantile of the non-conformity score differs significantly. We use the discovered

grouping in this prior step as an input to a group-conditional split conformal approach which now holds local conditional guarantees on the identified groups. Section 5 presents an implementation of this approach based on decision trees, which provide an interpretable clustering of the input space based on the input features (or an interpretable embedding of the same).

# 3 RELATED WORK

**Adaptive Conformal Sets.** Input-conditional coverage guarantees with finite samples are impossible without infinite width intervals Vovk [2012], Lei and Wasserman [2014]. However, an extensive line of work has focused on providing adaptive conformal sets that can capture heteroskedastic uncertainty in the model's prediction Romano et al. [2019], Kivaranovic et al. [2020]. Some works up-weight the non-conformity scores of calibration samples based on some distance notion to the test instance Mao et al. [2022], Guan [2023], Ghosh et al. [2023] or make assumptions on the data distribution Lei and Wasserman [2014], Barber et al. [2023]. These approaches do not integrate information about the non-conformity score in the weighting process. In contrast, approaches such as Han et al. [2022], Jung et al. [2022], Amoukou and Brunel [2023] model some statistic of the (conditional) non-conformity score distribution to re-weight, correct or learn the quantile threshold.

**Local Conditional Coverage.** Some works have proposed split conformal prediction methods for a predefined set of groups. For non-overlapping groups Mondrian conformal prediction provides finite sample guarantees Vovk et al. [2003], Vovk [2012]. The assumption here is that the observations in each group of the partition are exchangeable. For overlapping groups Foygel Barber et al. [2021] provides a conservative approach with finite sample guarantees (largest set from the groups that contain the test point). The work by Jung et al. [2022] learns the non-conformity score threshold conditioned on each group via quantile regression. Their approach has asymptotic guarantees, while Gibbs et al. [2023] proposed an alternative with finite sample guarantees.

**Group Identification for Local Conformal Prediction.** Lei and Wasserman [2014] proposes a "sandwich slicer" approach that bins the input features before applying a group/local conditional conformal approach, while Sesia and Romano [2021] proposes histogram binning of the ML model's output values. These approaches are simple but greedy, and do not leverage the information of the distribution of the non-conformity scores. Existing kernel-based localizers for conformal prediction Guan [2023], Han et al. [2022] do not partition the input space and do not integrate information about the non-conformity scores. The work by Amoukou and Brunel [2023] is the closest to our approach. They propose an adaptive conformal prediction approach that learns the non-conformity score weights with a random

forest that approximates a statistic of the non-conformity scores. To achieve this objective, they use a quantile random forest that approximates the distribution of the non-conformity scores on the calibration dataset. Moreover, they provide an approach to approximate the forest's weights with a partition function using a graph clustering method based on Louvain-Leiden Traag et al. [2019] with Markov Stability Delvenne et al. [2010]. Therefore, we compare against two of their variants. It is important to note that the quantile random forest algorithm by Meinshausen and Ridgeway [2006] does not minimize (an approximation of) the quantile objective $(1 - \alpha)$ but instead it minimizes the inter-leaf variance of the non-conformity scores; the leaves of this QRF algorithm store the entire list of non-conformity scores of train samples falling in the leave, rather than a single summary statistic. In our formulation, the learned partition function approximates the $1 - \alpha$ quantile of the non-conformity score, since we minimize pinball loss.

# 4 REGION IDENTIFICATION BASED ON NON-CONFORMITY SCORE QUANTILES

Given a non-conformity score, we want to discover regions in the input space that maximizes intra-group homogeneity of the score distribution, but still differ significantly between groups. These regions, if interpretable, provide useful insights about the uncertainty of a model's prediction. Moreover, they can be leveraged on different steps in the ML life cycle such as data filtering and collection.

Given a mis-coverage objective $\alpha$ we want to learn a mapping $\tau : \mathcal{X} \rightarrow \mathcal{G} \times \mathbb{R}$,[5] that outputs a computationally-identifiable set of groups and an estimate of the $1 - \alpha$ conformity score quantile for each group, $\tau(X) = (g_\tau(X), q_\tau(X))$. We use $g_\tau(X)$ to denote the group label and $q_\tau(X)$ to denote the corresponding quantile estimate (i.e., score threshold). We consider $\tau$ to belong to a family of piece-wise constant models $\mathcal{T}$ such that $\forall \tau \in \mathcal{T}, \forall x_1, x_2 \in \mathcal{X} : g_\tau(x_1) = g_\tau(x_2) \rightarrow q_\tau(x_1) = q_\tau(x_2)$.

Piece-wise constant models provide an interpretable characterization of the identified groups based on the input features, this is especially true for models such as trees, where the decision rules used to identify each group (leaf node) are clearly laid out. Note that our approach could also be applied to some interpretable feature space of the input by choosing $\tau(\phi(X))$ where $\phi(\cdot)$ is some mapping into an interpretable feature space. In particular, $\phi(X) = (X, f(X))$ makes the partitioning depend directly on the output of $f$. This allows the implicit identification of different uncertainty regions based on the model's prediction.

---

[5]we can also consider soft-clustering such that $\tau : \mathcal{X} \rightarrow \Delta^{|\mathcal{G}|-1} \times \mathbb{R}$

## 4.1 GENERALIZATION OF WORST GROUP MIS-COVERAGE

We want to learn a partition function $\tau(\cdot) \in \mathcal{T}$ that approximates the conditional quantile $F_{S|X}^{-1}(1-\alpha)$[6]. In practice, we have access to a finite dataset $\mathcal{D}$, on which the model family $\mathcal{T}$ may be prone to overfitting. Therefore, we want to choose a regularization parameter for $\mathcal{T}$ that ensures that the generalization properties of the final model are acceptable. In particular, we want to learn a partition where the worst group conditional coverage for the identified groups is as close as possible to $1-\alpha$. To do so, we first introduce our definitions of group conditional mis-coverage (Definition 4.1), worst group mis-coverage ratio (Definition 4.2), and then our proposed objective.

**Definition 4.1.** Consider a distance function $d : \mathbb{R} \times \mathbb{R} \to \mathbb{R}_{\geq 0}$, $\mathcal{G}$ a set of groups with membership function $g : \mathcal{X} \to \mathcal{G}$, a threshold $q \in \mathbb{R}$, and a target coverage $1-\alpha$. The group conditional mis-coverage of threshold function $q : \mathcal{X} \to \mathbb{R}$ over variable $S$ for a group $g_j \in \mathcal{G}$ based on distance $d$ is

$$MC_\alpha(q, g; g_j) = \mathbb{E}_{X,S}[d(1-\alpha, P(S \leq q(X)))|g(X) = g_j] \quad (6)$$

Following Definition 4.1, we are interested in measuring the worst group conditional mis-coverage w.r.t. the marginal baseline, that is, the model that outputs a single quantile estimate for the entire input space. This indicates if the proposed grouping, and corresponding quantile estimates, provide a significant improvement in terms of worst-group coverage over a simple, marginal approach. Definition 4.2 presents the proposed worst group mis-coverage ratio.

**Definition 4.2.** Consider a distance function $d : \mathbb{R} \times \mathbb{R} \to \mathbb{R}_{\geq 0}$, $\mathcal{G}_\tau$ the set of groups identified by $\tau(\cdot)$, $g_\tau(\cdot)$, the corresponding quantile estimator $q_\tau(\cdot)$, and $\hat{q} \simeq F_S^{-1}(1-\alpha)$ an empirical estimate of the average $1-\alpha$ quantile of $S$. Then, we define the worst mis-coverage ratio as

$$\text{MCR}_\alpha(\tau) = \frac{\max\limits_{g_j \in \mathcal{G}_\tau} MC_\alpha(q_\tau, g_\tau; g_j)}{\max\limits_{g_j \in \mathcal{G}_\tau} MC_\alpha(\hat{q}, g_\tau; g_j)} \quad (7)$$

For the distance function we consider $d(1-\alpha, p) = |1-\alpha - p|$ or $d(1-\alpha, p) = (1-\alpha - p)_+$, where the latter only considers under-coverage violations. The $\text{MCR}_\alpha(\tau)$ is less than 1 if the worst group mis-coverage on the proposed partition $\mathcal{G}_\tau$ is lower (better) than the worst mis-coverage of a single quantile estimate. In such case we may prefer the proposed partition over the baseline model.

Given two different group partitions, MCR allows us to compare which of the two partitions identified a set of groups that would be most benefited (in the worst group sense) by

---

[6] $F_{S|X}^{-1}(1-\alpha) = \inf\{\hat{s} \in supp(P_{S|X}) : P(S \leq \hat{S}|X) \geq 1-\alpha\}$

the new model over a marginal quantile estimate. Note that we cannot directly compare the worst group mis-coverage (MC) between two models directly, since the MCs are computed across different group definitions. The MCR uses the marginal baseline as an intermediary model, and allows us to compare these two models. The MCR ratio serves a similar role to the $R^2$ coefficient of determination (which compares the residual variance of a model against a constant baseline), but MCR is defined in terms of a pessimistic, worst-case scenario. MCR serves as a computationally efficient alternative to a full auditing approach where an auditor uses a sophisticated optimization procedure to identify the worst computationally identifiable group in terms of mis-coverage.

In practice, we observe that the proposed MCR is a better criteria for model selection and group identification than average pinball loss or simply worst group mis-coverage on a held out dataset. As we show in Section 6, selecting a model based only on average pinball loss on a held-out dataset tends to favor models with smaller group sizes whose quantile estimates later fail to generalize, with worst-group coverages that fall behind even the marginal quantile estimate. On the other hand, choosing only based on worst group mis-coverage (i.e. worst group MC instead of MCR) tends to discard groups of low probability even in the large sample regime. This is analized further in Section 6.

## 4.2 GROUP DISCOVERY OBJECTIVE

We want to learn a generalizable partition function $\tau(\cdot) \in \mathcal{T}$ that provides the best approximation of the conditional quantile $F_{S|X}^{-1}(1-\alpha)$. Additionally, we want to ensure that the worst group mis-coverage across the learned partition improves over the one achieved with a baseline model over the same partition. To do this, we consider a regularization function $\mathcal{R}_\theta(\tau)$ with parameters $\theta \in \Theta$ that controls the complexity of model $\tau(\cdot)$, the strength of the regularization function is chosen based on the empirical MCR score over a finite dataset $\mathcal{D}^a$. This is shown below

$$\theta^* \in \arg\min_{\theta \in \Theta} \text{MCR}_\alpha(\tau_\theta; \mathcal{D}^a)$$
$$s.t. : \tau_\theta \in \arg\min_{\tau \in \mathcal{T}} \mathbb{E}_{\mathcal{D}^b}\left[\ell_{1-\alpha}(q_\tau(X), S)\right] + \mathcal{R}_\theta(\tau). \quad (8)$$

The final partition function $\tau^*$ is the one that minimizes the empirical expected pinball loss with regularization $\mathcal{R}_{\theta^*}$,

$$\tau^* \in \arg\min_{\tau \in \mathcal{T}} \mathbb{E}_{\mathcal{D}^b}\left[\ell_{1-\alpha}(q_\tau(X), S)\right] + \mathcal{R}_{\theta^*}(\tau). \quad (9)$$

Note that the average pinball loss is estimated over a dataset $\mathcal{D}^b$ that is independent from $\mathcal{D}^a$ but sampled from the same distribution. The objective we propose in Eq. 8 essentially chooses the best model in terms of MCR score among the set of regularized, pinball-loss-minimizing models.

We stress that this objective is meaningful as a finite sample generalization constraint, since, given access to a sufficiently

large sample set to learn the group-conditional quantiles, the MCR would be zero. In essence, given sufficient samples, any quantile estimated for any partition of the input space would also have sufficient samples such that the estimated quantile would achieve near-exact group conditional coverage. An algorithm to achieve Eq. (8), and a formalization of the above statement are provided in the following section.

# 5 DISCOVERING AND CONFORMALIZING GROUPS IN PRACTICE

We consider $\theta$ to be a regularization parameter that is monotonically decreasing with model complexity. In this setting we propose Algorithm 1 to find the regularization strength $\theta^*$ that recovers the pinball loss minimizer with lowest MCR from a family of clustering methods $\mathcal{T}$. Following this discovery step, we then run a group-conditional conformal prediction mechanism on the discovered regions to conformalize the score quantiles and produce conformal sets/intervals with local coverage guarantees.

Proposition 5.1 shows that Algorithm 1 is optimal in the infinite sample regime, where the generalization objective is easily achieved by any partition. That is, even in the absence of generalization issues, Algorithm 1 correctly approximates the conditional quantile $F_{S|X}^{-1}(1-\alpha)$ within the desired model class (and finds the best pinball loss minimizer in the presence of generalization challenges otherwise). Although this particular result hinges on the 'infinite sample' assumption, we stress that Algorithm 1 also performs group-conditional conformal predictions on each of the recovered groups (last step in Algorithm 1) which does have finite sample group conditional guarantees as shown in Eq. 12

**Proposition 5.1.** *Given the objective in Eq. 8, if* $\mathcal{D}^a = P(X, S)$ *(infinite sample regime) and* $\theta_0$ *in Algorithm 1 is the weakest admissible regularization, then* $\tau^* = \tau_{\theta_0}$, *which also minimizes pinball loss over all admissible regularizations* $\mathbb{E}_{\mathcal{D}}\big[\ell_{1-\alpha}(q_{\tau^*}(X), S))\big] \leq \mathbb{E}_{\mathcal{D}}\big[\ell_{1-\alpha}(q_{\tau_\theta}(X), S))\big], \forall \theta \in \Theta$ *such that* $\theta \geq \theta_0$.

**Learning Generalizable Quantile Score Regions.** Algorithm 1 assumes access to a solver for the $\tau_\theta$ objective, denoted as $\mathcal{M}_{1-\alpha}$, and a conformal prediction mechanism $\mathcal{A}_{CP}$ in addition to a dataset ($\mathcal{D}_1$) containing input samples and their corresponding non-conformity scores. The initial parameter $\theta_0$ is the weakest acceptable regularization due to interpretability purposes (e.g., maximum tree depth), $\prod_\Theta(\cdot)$ a projection operator into the regularization parameter space, and $\Delta_\theta > 0$ a step size that guarantees a change in $\theta_t$ when projected into $\Theta$ unless the minimum admissible complexity bound has been reached. The final clustering model $\tau^*(\cdot) = (g_{\tau^*}, q_{\tau^*})(\cdot)$ is learned using the best regularization parameter $\theta^* \in \Theta$ in terms of MCR. This simple approach of steadily increasing regularization strength in

finite increments $\delta_\theta$ and stopping when MCR fails to improve is sufficient for our purposes, but more sophisticated zero-order approaches could substitute this update strategy.

**Conformalizing the Conditional Quantiles of the Discovered Regions.** The learned clustering function $g_{\tau^*}(\cdot)$ is then fed into a group conditional conformal prediction mechanism, $\mathcal{A}_{CP}$ such as Vovk [2012], Foygel Barber et al. [2021], Jung et al. [2022], Gibbs et al. [2023] to provide conformalized thresholds for each identified group.

For example, for clustering functions $g_{\tau^*}(\cdot)$ that partition the space with no overlaps, we consider a standard group conditional split conformal method where $\mathcal{A}_{CP}$ provides the conformal quantile estimator $q_{\tau_{CP}^*}(\cdot)$ based on the conformal quantile of each identified group. The corresponding conformal set $C_\tau(X_{n+1})$ for a new sample is defined as:

$$C_\tau(X_{n+1}) = \{y \in \mathcal{Y} : S_f(X_{n+1}, y) \leq q_{\tau_{cp}}(X_{n+1})\} \tag{10}$$

where the conformal quantile function $q_{\tau_{cp}}(\cdot)$ is

$$q_{\tau_{cp}}(X_{n+1}) = \mathbb{Q}_{1-\alpha}\Big(\sum_{i=1}^n \frac{\mathbf{1}[g_{n+1} = g_i]}{n_{g_i} + 1}\delta_{s_i} + \frac{1}{n_{g_{n+1}} + 1}\delta_\infty\Big) \tag{11}$$

with $g_{\tau^*}(x_i) = g_i$, and $n_{g_i}$,[7] the number of samples of group $g_i$ in dataset $\mathcal{D}_2, \forall i \in [n + 1]$

Moreover, for each identified group $g \in \mathcal{G}_{\tau^*}$ the coverage guarantees become

$$1 - \alpha \leq P\Big(Y_{n+1} \in C_\tau(X_{n+1}) | g_{\tau^*}(X_{n+1}) = g\Big) \leq 1 - \alpha + \frac{1}{n_g + 1}. \tag{12}$$

Note that the upper bound depends on the number of samples $n_g$ of group $g$ in the calibration set.

## 5.1 LEARNING DECISION-TREE-BASED REGIONS

Decision trees make a natural candidate for learning partition functions, since they are inherently interpretable, especially at lower tree depths. We need access to a solver $M_{1-\alpha}$ that, given a dataset and a regularization parameter, provides a tree that minimizes the $1 - \alpha$ average pinball loss as in Eq.8. The challenge we face with existing decision tree regression optimizers is that, as far as we know, available solvers do not support pinball loss. Therefore, we first train a surrogate model $h^* \in \mathcal{H}$ that does have access to pinball loss solvers. Then, we approximate the output of $h^*$ with the decision tree by minimizing the mean square error loss against the surrogate model's predicted (input dependent) quantile. The procedure described here to learn a decision tree for pinball loss minimization is summarized in

---

[7] $n_{g_i} = |\{j : g_{\tau^*}(x_j) = g_i\}_{i \in \mathcal{D}_2}|$

**Algorithm 1** Region Identification Meta-Algorithm

**Require:** i.i.d. dataset $\mathcal{D}_1$ of input samples and corresponding non-conformity scores. $M_{1-\alpha}(\cdot,\cdot):\mathcal{D}\times\Theta\to\mathcal{T}$ solver for $\tau$ in Eq. 8. $\mathcal{A}_{CP}:\mathcal{D}\times\mathcal{G}^{|\mathcal{D}|}\to\mathbb{R}^{|\mathcal{G}|}$ group-conditional prediction mechanism. $\theta_0\in\Theta$ weakest acceptable regularization parameter, $\Delta_\theta$ regularization step size.
  // Region Identification
  $MCR^* \leftarrow \infty$ Initialize best MCR init
  **for** $t = 0, \ldots, T$ **do**
    $MCR_t \leftarrow \{\}$ Initialize MCR set $t$
    // K-fold Cross validation
    **for** $k = 1, \ldots, K$ **do**
      Split $\mathcal{D}_1$ randomly into $\mathcal{D}^{a,k}$ and $\mathcal{D}^{b,k}$
      $\tau_\theta = M_{1-\alpha}(\mathcal{D}^{b,k}, \theta_t)$,
      $MCR_t \leftarrow MCR_t \cup MCR(\tau_\theta, \mathcal{D}^{a,k})$
    **end for**
    $sMCR = mean(MCR_t) + std(MCR_t)$
    **if** $sMCR < MCR^*$ **then**
      $MCR^* \leftarrow sMCR$, $\theta^* \leftarrow \theta_t$
    **end if**
    $\theta_{t+1} \leftarrow \prod_\Theta(\theta_t + \Delta_\theta)$
  **end for**
  $\tau^* \leftarrow M_{1-\alpha}(\mathcal{D}_1, \theta^*)$
  // Conformalize group conditional quantile predictor
  $q_{\tau_{cp}} \leftarrow \mathcal{A}_{CP}(\mathcal{D}_1, \{g_{\tau^*}(x_i)\}_{i\in\mathcal{D}_1})$

**output** $\tau_{cp} = (q_{\tau_{cp}}, g_{\tau^*})$

the following objective

$$\tau_\theta \in \arg\min_{\tau\in\mathcal{T}} \mathbb{E}_{\mathcal{D}^b}\left[(q_\tau(X) - h^*(X))^2\right] + \mathcal{R}_\theta(\tau),$$
$$s.t. \quad h^* \in \arg\min_{h\in\mathcal{H}} \mathbb{E}_{\mathcal{D}^b}\left[\ell_{1-\alpha}(h(X), S))\right]. \quad (13)$$

In our experiments, we take $\mathcal{H}$ to be a family of gradient boosting decision trees that support pinball loss Ke et al. [2017], and use hyperparameter optimization Akiba et al. [2019] to minimize overfitting in the surrogate model $h^*$.

# 6 EXPERIMENTS

We evaluate the proposed method on a variety of datasets and show how the proposed MCR-score-based method is able to identify a set of groups whose local coverage is close to the desired target, and show that this diminishes the under- and over-coverage gaps compared to the alternatives.

## 6.1 REGRESSION DATASET RESULTS

We used Gradient boosting (LGBM) Ke et al. [2017] as our base regressor $f$; the hyperparameters for each dataset were selected using hyperparameter optimization using Akiba et al. [2019] to minimize validation loss. Additional results using Lasso are shown in Appendix A.3. For all experiments, we split the available training data as follows: 40% train, 40% calibration, 20% test. We use a target coverage/validity of 0.9 (90%, $\alpha = 0.1$).

| model | MCR | coverage | | | num groups |
|---|---|---|---|---|---|
| | | average | max group | min group | |
| **Housing: nsamples = 506, nfeatures = 13 \| LGBM-Regressor R2 = 0.64 ± 0.03** | | | | | |
| LCP-RF-G | 1.45± 1.14 | .8± .04 | .91± .07 | .64± .15 | 3.6± .55 |
| RF-G | .77± .6 | .93± .03 | .99± .01 | .86± .06 | 3.6± .55 |
| PB-KMEANS | .81± .3 | .92± .02 | .97± .04 | .68± .33 | 8.4± 8.65 |
| MCR-KMEANS | .75± .12 | **.91± .05** | **.95± .05** | .84± .13 | 2.2± 1.64 |
| PB_DTREE | .68± .31 | .89± .02 | .94± .03 | .83± .04 | 3.4± .55 |
| MCR_DTREE | **.65± .17** | .92± .03 | **.95± .04** | **.88± .07** | 2.2± 1.3 |
| **Concrete: nsamples = 1030, nfeatures = 8 \| LGBM-Regressor R2 = 0.82 ± 0.026** | | | | | |
| LCP-RF-G | 1.84± 1.66 | .83± .01 | .94± .05 | .69± .11 | 4.6± .55 |
| RF-G | .82± .68 | **.9± .05** | .97± .02 | .81± .11 | 4.6± .55 |
| PB-KMEANS | .66± .48 | .91± .05 | .97± .05 | .83± .07 | 7.0± 3.24 |
| MCR-KMEANS | .88± .27 | .91± .05 | **.92± .06** | **.88± .05** | 4.2± 7.16 |
| PB_DTREE | .94± .57 | .89± .04 | .98± .02 | .77± .07 | 6.6± .55 |
| MCR_DTREE | **.55± .72** | .9± .04 | **.92± .06** | **.88± .04** | 2.4± 2.61 |
| **Energy: nsamples = 768, nfeatures = 8 \| LGBM-Regressor R2 = 0.93 ± 0.05** | | | | | |
| LCP-RF-G | .99± 1.31 | .87± .06 | .97± .03 | .65± 0.05 | 5.0± 1.0 |
| RF-G | .65± .1 | **.92± .03** | .99± .02 | 0.87± .06 | 4.8± 1.64 |
| PB-KMEANS | 1.04± .34 | .85± .07 | 1.0± .0 | .07± .15 | 47.8± 1.79 |
| MCR-KMEANS | .68± .3 | .94± .03 | **.96± .05** | .78± .17 | 1.6± 9.5 |
| PB_DTREE | .63± .5 | .93± .03 | .97± .02 | .87± .07 | 3.6± 1.52 |
| MCR_DTREE | **.5± .46** | **.92± .03** | **.96± .03** | **.88± .07** | 3.2± 1.64 |
| **Power: nsamples = 9568, nfeatures = 4 \| LGBM-Regressor R2 = 0.95 ± 0.01** | | | | | |
| LCP-RF-G | 3.67± 2.26 | .82± .05 | .86± .03 | .78± .07 | 4.4± 1.95 |
| RF-G | **.47± .22** | **.9± .0** | **.92± .01** | **.88± .02** | 5.0± .71 |
| PB-KMEANS | .76± .18 | .9± .01 | .95± .03 | .85± .02 | 15.0± 7.55 |
| MCR-KMEANS | .66± .23 | .91± .01 | .96± .03 | .86± .02 | 16.6± 10.26 |
| PB_DTREE | 1.13± .6 | .9± .0 | .98± .04 | .76± .09 | 17.2± 9.26 |
| MCR_DTREE | .57± .2 | **.9± .01** | **.92± .03** | **.88± .03** | 5.8± 8.56 |
| **Protein: : nsamples = 45730, nfeatures = 9 \| LGBM-Regressor R2 = 0.46 ± 0.04** | | | | | |
| LCP-RF-G | .83± .56 | **.9± .01** | .94± .05 | .85± .02 | 10.5± 6.4 |
| RF-G | .61± .36 | **.9± .0** | .95± .05 | .88± .03 | 11.0± 7.0 |
| PB-KMEANS | .59± .57 | **.9± .0** | 1.0± .0 | .71± .22 | 4.8± 5.67 |
| MCR-KMEANS | .47± .3 | **.9± .0** | .97± .05 | .87± .03 | 11.4± 8.26 |
| PB_DTREE | .79± .27 | **.9± .0** | 1.0± .0 | .81± .01 | 31.2± .45 |
| MCR_DTREE | **.17± .14** | **.9± .0** | **.91± .01** | **.89± .01** | 4.4± .89 |
| **kin8mn: : nsamples = 8192, nfeatures = 8 \| LGBM-Regressor R2 = 0.62 ± 0.03** | | | | | |
| LCP-RF-G | 2.32± 1.1 | .8± .02 | .84± .02 | .75± .04 | 4.6± 1.34 |
| RF-G | **.32± .18** | **.9± .0** | .93± .01 | .87± .01 | 5.2± .45 |
| PB-KMEANS | .96± 0.67 | .92± .0 | 1.0± .0 | .72± .03 | 41.0± 8.57 |
| MCR-KMEANS | .76± .16 | .91± .02 | .94± .05 | .82± .11 | 20.6± 7.06 |
| PB_DTREE | .73± .39 | .9± .01 | .97± .03 | .8± .07 | 16.4± 6.58 |
| MCR_DTREE | .4± .2 | **.9± .01** | **.91± .02** | **.89± .02** | 3.0± 1.41 |

Table 1: Comparison between the group discovery partition methods. We show MCR, marginal, minimum, and maximum coverage group coverage on the identified partition. We also report the number of groups per approach. Standard deviations are computed across 5 data splits. The proposed MCR_DTREE is consistently better in terms of MCR, with values consistently below 1, indicating that the discovered groups improve worst-group under-coverage w.r.t. to single threshold SCP. Every dataset uses a LGBM regressor as the base model. We highlight the lowest MCR and the smallest average coverage above the objective (0.9) since models with larger coverages are less efficient. For methods that achieved the marginal coverage objective we highlight the max and min group coverage closest to the 0.9 objective.

**Datasets.** We considered six regression tasks based on datasets from the UCI repository Asuncion and Newman [2007]. These are the Boston Housing price prediction (14 attributes, Housing) Harrison Jr and Rubinfeld [1978]; Energy efficiency prediction (12 building parameters, Energy) Tsanas and Xifara [2012]; Concrete compressive strength

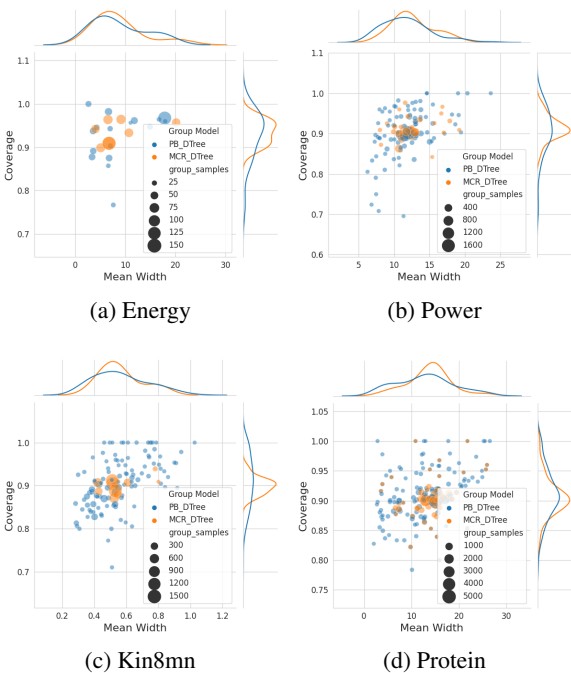

(a) Energy

(b) Power

(c) Kin8mn

(d) Protein

Figure 2: Scatter and distribution plot of the prediction interval widths (x-axis) versus coverage (y-axis) of the groups discovered by the proposed MCR_DTREE and PB_DTREE methods across 6 datasets. We plot all the groups obtained across 5-Fold realizations. The size of the group's points represents the group size. The target coverage is 0.9, we observe that MCR_DTREE tends to identify a smaller number of groups of varying sizes, with group-conditional coverages concentrated around the 0.9 objective. Moreover, the identified groups show diversity in the range of interval widths. PB_DTREE detects a significant larger number of (smaller) groups, with a larger variance in terms of group-conditional coverage. Additional plots in Appendix A.3.

prediction (8 attributes, Concrete) Yeh [2007]; Estimation of the size of the residue based on different physical and chemical properties of protein tertiary structure (Protein) Rana [2013] ; Net hourly electrical energy output prediction of a combined cycle power plant (4 features, Power) Tfekci and Kaya [2014]; Predict the distance of the end-effector from a target based on the forward kinematics of a robot arm (kin8mn) Rasmussen et al. [1996], Corke [1996].

**Methods.** We evaluate the performance of Algorithm 1 choosing $\tau$ to be a decision tree that minimizes the pinball loss as described in Section 5.1. We use standard group-conditional split conformal ($\mathcal{A}_{CP}$) Vovk [2012] and denote the final model as MCR_DTREE. For the MCR score (Eq. 7) we selected $d(1 - \alpha, p) = (1 - \alpha - p)_+$ as our under-coverage distance function. We constrain our decision trees

to a minimum of 50 samples per leaf and max depth of 5. We set the cost complexity pruning variable as the regularization parameter $\theta$ with $\theta_0 = 1e - 5$ and $\Delta_{\theta_t} = 9 \times \theta_t$. We compare against a decision tree that minimizes average pinball loss (i.e., Algorithm 1 where MCR is replaced by average pinball loss), we denote it as PB_DTREE. Additionally, we compare against the group-wise random forest localizer conformalization method (LCP-RF-G) proposed by Amoukou and Brunel [2023] which generates a partition using conformity score weights extracted from a random forest, and later use a standard split conformal approach based on their identified groups (RF-G). Finally, we examine a simple K-means clustering in the input space, where the number of clusters is chosen based on best average pinball loss (PB-KMEANS) and best MCR (MCR-KMEANS) with cross validation.

**Coverage on Identified Groups.** Table 1 shows the minimum and maximum group coverage for the partitions recovered by each approach. We observe that the proposed MCR_DTREE identifies partitions that consistently provide the best (or second best) minimum coverage, and smallest gap between maximum and minimum group coverage, all while achieving the target marginal coverage of 0.9. In general, MCR_DTREE tends to identify a smaller set of groups, with a wide range of interval widths as shown in Figure 2. Moreover, it achieves the smallest MCR when compared to the competing baselines. The MCR of MCR_DTREE is consistently below 1, indicating that a baseline SCP approach would yield worse results in terms of worst group under-coverage. We note that the partition identified by RF-G, once integrated with split conformal prediction, has significantly better performance than their LCP-RF-G alternative. RF-G achieves comparable results in some of the datasets, with larger disparity in terms of coverage gap between the identified groups, and worse MCR. PB-KMEANS and MCR-KMEANS have large variances in their performance, potentially due to the fact that KMEANS clusters do not leverage the non-conformity scores.

**Size and Efficiency of the Identified Groups.** Figure 2 shows the joint distribution of the mean width and coverage of the identified groups by MCR_DTREE and PB_DTREE approaches across all datasets. We observe that MCR_DTREE tends to identify a smaller number of groups when compared to PB_DTREE. PB_DTREE tends to identify multiple groups of small sizes, with a wide range of widths and coverage ranges. MCR_DTREE is able to identify groups with diverse widths (as we can see in the marginal distribution of the mean width) but the identified groups have coverages closer to the desired objective of 0.9.

**Interpretable Groups.** Figure 3 in Appendix A.3 shows the trees discovered by MCR_DTREE. The discovered groups have different interval widths, indicating that the uncertainty on the model's prediction is non-uniform across

the input space. Moreover, groups with higher uncertainty (larger mean width) tend to have a smaller size. This can inform a data collection process by encouraging the collection of samples from the identified high uncertainty minorities.

# 7 CONCLUSION

Here we propose a method to learn an interpretable partition of the input space based on the uncertainty of a black box model's prediction. We leverage the conformal prediction framework and decision tree models to identify a set of groups of varying sizes where the quantile of the nonconformity scores are as homogeneous as possible within the group but significantly different across different groups. We propose a fitness criteria (group miscoverage ratio, MCR) and accompanying algorithm to achieve this and show its effectiveness in a varying set of regression datasets. Our proposed method is able to discover a set of groups with better local coverage performance than competing methods.

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

# Identifying Homogeneous and Interpretable Groups for Conformal Prediction (Supplementary Material)

**Natalia Martinez Gil**[1]     **Dhaval Patel**[1]     **Chandra Reddy**[1]     **Giridhar Ganapavarapu**[1]     **Roman Vaculin**[1]

**Jayant Kalagnanam**[1]

[1]IBM Research, Yorktown Heights, New York, USA

## A    APPENDIX

### A.1    PROOFS

Restatement of Proposition 5.1

**Proposition A.1.** *Given the objective in Eq. 8, if $\mathcal{D}^1 = P(X, S)$ (infinite sample regime) and $\theta_0$ in Algorithm 1 is the weakest admissible regularization, then $\tau^* = \tau_{\theta_0}$, which also minimizes pinball loss over all admissible regularizations* $\mathbb{E}_{\mathcal{D}}\big[\ell_{1-\alpha}(q_{\tau^*}(X), S))\big] \leq \mathbb{E}_{\mathcal{D}}\big[\ell_{1-\alpha}(q_{\tau_\theta}(X), S))\big], \forall \theta \in \Theta$ *such that $\theta \geq \theta_0$.*

*Proof.* We first show that in the infinite sample regime the MCR is zero $\forall \theta \in \Theta$, making all $\theta$ equivalent according to the MCR criteria. Then we show that Algorithm 1 would choose $\theta^* = \theta_0$ and since $\theta_0$ is the lowest regularization it achieves the smallest expected pinball loss.

Given access to the real distribution $\mathcal{D}_1 = P(X, S)$ for any $\theta \in \Theta$ we get a finite set partition $\mathcal{G}_{\tau_\theta}$ such that the $1 - \alpha$ quantile estimate $q_{\tau_\theta}(X)$ is the exact group conditional quantile of the non-conformity score distribution for the group that contains the instance $X$.

$$q_{\tau_\theta}(X) = F^{-1}_{S|G=g_{\tau_\theta}(X)}(1 - \alpha) \tag{14}$$

where $g_{\tau_\theta}(X) \in \mathcal{G}_{\tau_\theta}, \forall X \in \mathcal{X}$. Then, in this asymptotic regime the group conditional miscoverage (Definition 4.1) $MC_\alpha(q_{\tau_\theta}, g_{\tau_\theta}; g_j) = 0 \ \forall g \in \mathcal{G}_{\tau_\theta}$ , $\forall g \in \mathcal{G}_{\tau_\theta}$ and $\forall \theta \in \Theta$. Then MCR$_\alpha$ $(\tau_\theta)$ as defined in Eq. 7 is 0 $\forall \theta \in \Theta$.

Since Algorithm 1 terminates on the first $\theta$ that achieves the minimum MCR then $\theta^* = \theta_0$. Since $\theta_0$ is the weakest regularization, and we assume infinite sample regime to learn $\tau_\theta \forall \theta \in \Theta$ then $\mathbb{E}_{\mathcal{D}}\big[\ell_{1-\alpha}(q_{\tau^*}(X), S))\big] \leq \mathbb{E}_{\mathcal{D}}\big[\ell_{1-\alpha}(q_{\tau_\theta}(X), S))\big], \forall \theta \in \Theta$ such that $\theta \geq \theta_0$.

$\square$

### A.2    EXPERIMENTAL DETAILS

#### A.2.1    Learning Decision Tree Based Regions

We learn a decision tree that approximates the non-conformity score quantile by optimizing Eq. 13. To do so, we first learn a surrogate model $h$ that minimizes the pinball loss $1 - \alpha$ of the non-conformity scores.

**Step 1: Learn Surrogate Model** $h$    In our experiments $h$ is an LGBM quantile regressor that we learn using Optuna Akiba et al. [2019] with the following hyperparameters over 5-fold validation where the final set of parameters for $h^*$ is chosen based on best average pinball loss plus one standard deviation.

- Optimizer Configuration: N_TRIALS = 200, TIMEOUT = 11200.

- LGBM Model Parameters Exploration: LAMBDA_1 $\sim$ loguniform$(1e - 8, 10.0)$, LAMBDA_2 $\sim$ loguniform$(1e - 8, 10.0)$, LEARNING_RATE $\sim$ loguniform$(1e - 8, 10.0)$, bagging_fraction $\in [0.4, 1.0]$, bagging_freq $\in [1, 7]$, num_leaves $\in [2, 100]$, num_boost_round $\in [1, 100]$, min_child_samples $\in [50, 200]$, max_depth $= 2$ ,

**Step 2: Learn The Decision Tree Model** $\tau$    To learn $\tau$ we optimize the mean square error distance w.r.t. the prediction of the quantile LGBM regressor $h^*$ learned in the previous step. As stated in Section 6 in Algorithm 1 we consider trees up to a maximum depth of 5 and at least 50 samples per leaf. The regularization parameter $\theta$ is the cost complexity pruning variable. We set $\theta_0 = 1e - 5$ and a step size $\Delta_{\theta_t} = 9 \times \theta_t$.

## A.3    ADDITIONAL EXPERIMENTS

Figure 3b shows the decision trees that were obtain for the different datasets. We observe that the discovered regions have different prediction interval widths indicating that the model's prediction uncertainty is significantly different. Figure 4 shows the scatter and joint distribution between the prediction interval widths and coverage of the discovered groups. It extends Figure 2 in the main manuscript including all datasets and the groups discovered by the RF-G approach proposed by Amoukou and Brunel [2023]. Table 2 shows the same comparison presented in Table 1 but for a LASSO base model. We observe that the number of discovered groups by the proposed method MCR_DTREE is larger than those of a LGBM regression model for the same dataset (Table 1). In most cases, the LGBM model is equal or better than LASSO in terms of r2 score, and therefore reduces the unexplained variance of the target $Y|X$. This leads to less regions of different uncertainty and tighter prediction sets.

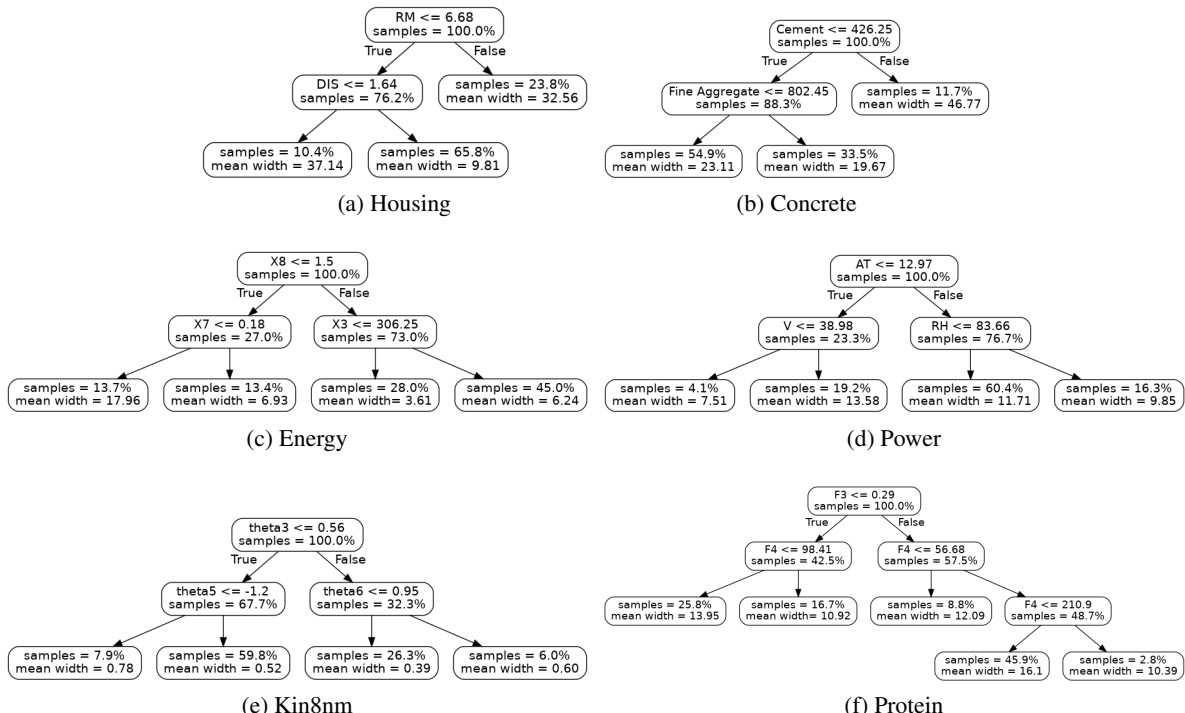

Figure 3: Example of decision trees identified for each regression dataset. (3a) In the Housing dataset groups are defined based on the features corresponding to average number of rooms per dwelling (RM) and weighted distances to five Boston employment centers (DIS). (3b) In the Concrete dataset the groups are defined based on the Cement and Fine Aggregate components ($kg$ in a $m^3$ mixture). (3c) the groups in the Energy dataset are defined based on Glazing Area Distribution (X8), Glazing Area (X7) and Wall Area (X3). (3d) In the Power dataset groups are defined based on Ambient Temperature (AT), Exhaust Vacuum (V) and Relative Humidity (RH). (3e) In the kin8nm dataset the groups are defined by the measurements on sensors from links 3, 5 and 6 from the robot arm. (3f) In the protein dataset the groups are defined by the features corresponding to fractional area of exposed non polar residue (F3) and fractional area of exposed non polar part of residue (F4).

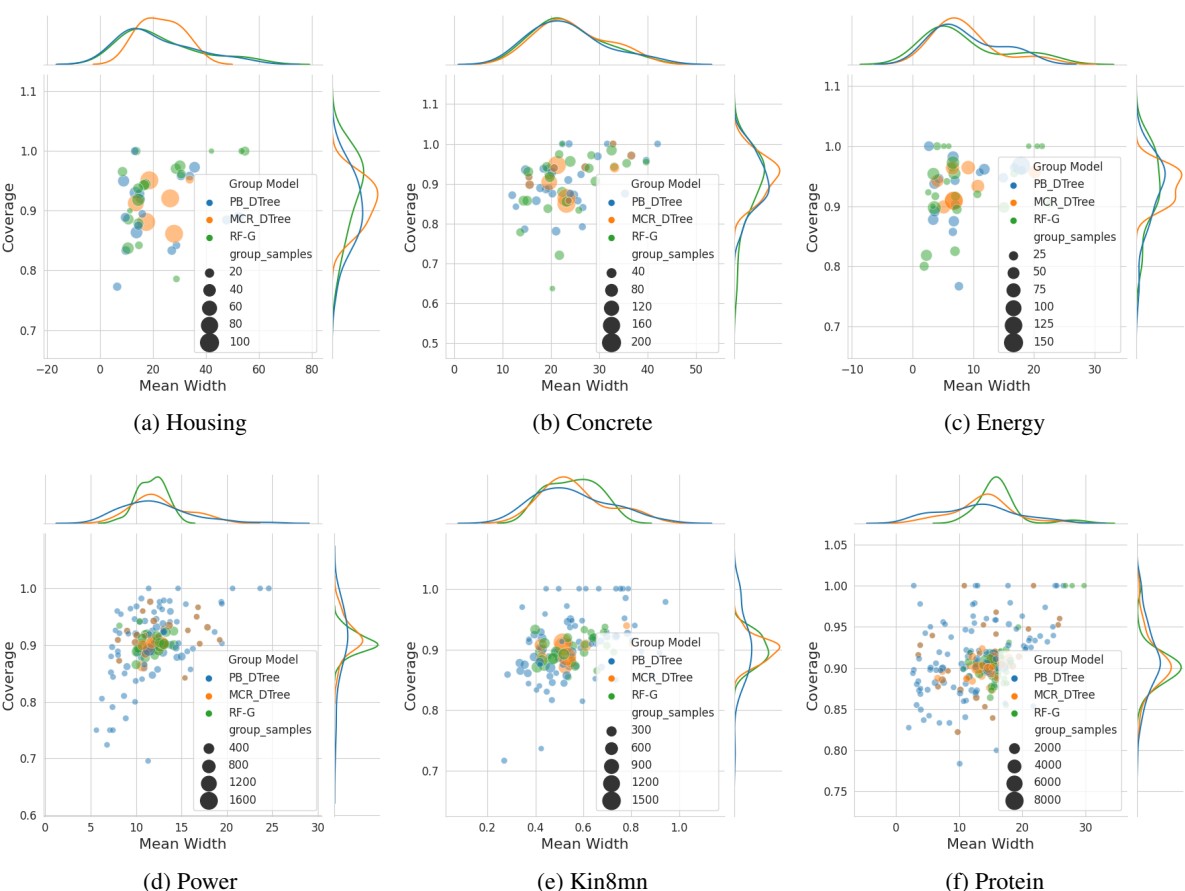

(a) Housing        (b) Concrete        (c) Energy

(d) Power        (e) Kin8mn        (f) Protein

Figure 4: Scatter and distribution plot of the prediction interval widths (x-axis) versus coverage (y-axis) of the groups discovered by the proposed MCR_DTREE, PB_DTREE and RF-G methods across 6 datasets. Here we plot all the groups obtained across 5-Fold realizations. The size of the groups points represents the group size (number of samples). The target coverage is 0.9, we observe that MCR_DTREE tends to identify a smaller number of groups of varying sizes, with group-conditional coverages concentrated around the 0.9 objective. Moreover, the identified groups show diversity in the range of interval widths. PB_DTREE detects a significant larger number of (smaller) groups, with a larger variance in terms of group-conditional coverage.

| model | MCR | coverage | | | num groups |
| | | average | max group | min group | |
|---|---|---|---|---|---|
| **Housing: nsamples = 506, nfeatures = 13 | LASSO-Regressor R2 = 0.69 $\pm$ 0.04** | | | | | |
| LCP-RF-G | 2.71±0.77 | 0.8±0.06 | 0.91±0.08 | 0.75±0.07 | 2.6±0.55 |
| RF-G | 0.42±0.38 | **0.91±0.03** | 0.96±0.03 | 0.81±0.15 | 3.2±0.45 |
| PB-KMEANS | 1.47±0.49 | 0.86±0.03 | 0.98±0.03 | 0.44±0.43 | 14.2±15.02 |
| MCR-KMEANS | 1.35±0.74 | 0.88±0.04 | 0.97±0.03 | 0.69±0.38 | 7.4±11.52 |
| PB_DTREE | 0.32±0.21 | 0.88±0.03 | 0.98±0.05 | 0.83±0.05 | 4.0±1.87 |
| MCR_DTREE | **0.25±0.39** | 0.89±0.04 | **0.95±0.04** | **0.84±0.07** | 3.6±2.07 |
| **Concrete: nsamples = 1030, nfeatures = 8 | LASSO-Regressor R2 = 0.60 $\pm$ 0.05** | | | | | |
| LCP-RF-G | 1.37±1.12 | 0.83±0.02 | **0.96±0.04** | 0.7±0.05 | 5.4±0.55 |
| RF-G | 0.29 ±0.15 | 0.91±0.02 | 0.98±0.03 | 0.8±0.08 | 5.0±0.71 |
| PB-KMEANS | 0.89±0.48 | **0.9±0.05** | 1.0±0.0 | 0.26±0.37 | 37.2±16.93 |
| MCR-KMEANS | 0.43±0.43 | 0.92±0.02 | 0.97±0.03 | 0.7±0.3 | 15.8±18.98 |
| PB_DTREE | 0.25±0.14 | **0.9±0.03** | 1.0±0.0 | 0.8±0.07 | 7.0±2.24 |
| MCR_DTREE | **0.15±0.09** | **0.9±0.03** | 1.0±0.0 | **0.84±0.04** | 6.8±2.39 |
| **Energy: nsamples = 768, nfeatures = 8 | LASSO-Regressor R2 = 0.91 $\pm$ 0.005** | | | | | |
| LCP-RF-G | 0.38±0.19 | 0.88±0.05 | **0.98±0.03** | 0.8±0.08 | 4.8±0.45 |
| RF-G | 0.12±0.12 | **0.94±0.02** | 1.0±0.0 | 0.87±0.06 | 5.0±0.71 |
| PB-KMEANS | 1.07±0.77 | 0.87±0.04 | 0.99±0.02 | 0.18±0.4 | 38.2±19.15 |
| MCR-KMEANS | 0.32±0.41 | **0.94±0.03** | 0.98±0.04 | 0.83±0.13 | 13.0±11.92 |
| PB_DTREE | 0.12±0.16 | **0.94±0.02** | 0.99±0.03 | 0.84±0.11 | 9.0±3.46 |
| MCR_DTREE | **0.05±0.09** | **0.94±0.02** | 0.98±0.02 | **0.89±0.03** | 6.0±3.24 |
| **Power: nsamples = 9568, nfeatures = 4 | LASSO-Regressor R2 = 0.93 $\pm$ 0.003** | | | | | |
| LCP-RF-G | 2.04±1.26 | 0.82±0.05 | 0.86±0.08 | 0.78±0.05 | 6.0±2.24 |
| RF-G | 0.83±0.57 | **0.9±0.0** | **0.93±0.02** | 0.87±0.01 | 5.2±0.84 |
| PB-KMEANS | 0.73±0.27 | 0.91±0.01 | 0.99±0.02 | 0.78±0.05 | 37.2±5.22 |
| MCR-KMEANS | 0.46±0.15 | **0.9±0.0** | **0.93±0.03** | **0.88±0.03** | 6.0±7.28 |
| PB_DTREE | 0.08±0.05 | **0.9±0.01** | 0.94±0.03 | 0.87±0.02 | 6.4±4.16 |
| MCR_DTREE | **0.06±0.05** | **0.9±0.0** | 0.94±0.01 | **0.88±0.02** | 7.4±3.71 |
| **Protein: : nsamples = 45730, nfeatures = 9 | LASSO-Regressor R2 = 0.28 $\pm$ 0.01** | | | | | |
| LCP-RF-G | 0.89±0.56 | 0.87±0.03 | **0.92±0.02** | 0.75±0.04 | 5.8±1.6 |
| RF-G | 0.44±0.37 | **0.9±0.0** | 0.95±0.05 | 0.87±0.02 | 6.00±1.59 |
| PB-KMEANS | 0.71±0.75 | **0.9±0.0** | 1.0±0.0 | 0.65±0.21 | 42.6±7.86 |
| MCR-KMEANS | 0.52±0.21 | **0.9±0.0** | 0.96±0.05 | 0.76±0.24 | 16.2±12.91 |
| PB_DTREE | 0.44±0.37 | **0.9±0.0** | 1.0±0.0 | 0.83±0.02 | 15.6±0.89 |
| MCR_DTREE | **0.2±0.08** | **0.9±0.0** | 0.93±0.03 | **0.89±0.01** | 5.6±2.19 |
| **kin8mn: : nsamples = 8192, nfeatures = 8 | LASSO-Regressor R2 = 0.40 $\pm$ 0.007** | | | | | |
| LCP-RF-G | 1.68±0.29 | 0.79±0.01 | 0.81±0.01 | 0.77±0.01 | 3.0±0.0 |
| RF-G | **0.21±0.04** | **0.9±0.01** | **0.91±0.01** | **0.88±0.0** | 3.2±0.45 |
| PB-KMEANS | 0.67±0.16 | 0.92±0.01 | 0.99±0.01 | 0.76±0.04 | 39.4±14.06 |
| MCR-KMEANS | 0.44±0.37 | **0.9±0.01** | 0.93±0.04 | 0.87±0.05 | 11.6±21.47 |
| PB_DTREE | 0.41±0.36 | 0.89±0.01 | 0.98±0.04 | 0.82±0.07 | 14.2±3.03 |
| MCR_DTREE | 0.24±0.18 | **0.9±0.01** | 0.94±0.04 | **0.88±0.02** | 6.4±5.37 |

Table 2: Comparison between the group discovery partition methods. We show MCR, marginal, minimum, and maximum coverage group coverage on the identified partition. We also report the number of groups per approach. Standard deviations are computed across 5 data splits. The proposed MCR_DTREE is consistently better in terms of MCR, with values consistently below 1, indicating that the discovered groups improve worst-group under-coverage w.r.t. to single threshold SCP. Every dataset uses a LASSO regressor as the base model. We highlight the lowest MCR and the smallest average coverage above the objective (0.9). For methods that achieved the marginal coverage objective we highlight the max and min group coverage closest to the 0.9 objective.