# OpenReview forum: "Identifying Homogeneous and Interpretable Groups for Conformal Prediction"
_auai.org/UAI/2024/Conference — UAI 2024 poster_

### Official Review · Reviewer_eP9C · 2024-03-08

**Q2-1 Originality-Novelty:** 2
**Q2-2 Correctness-Technical Quality:** 2
**Q2-5 Clarity Of Writing:** 3

**Q1 Summary And Contributions:**

The authors propose a clustering procedure that is meant to feed a Group-conditional conformal prediction algorithm. The complete procedure is then compared with existent approaches.

**Q2-3 Extent To Which Claims Are Supported By Evidence:**

2: Fair: the main claims are somewhat supported by evidence (but the experimental evaluation may be weak, or does not match entirely with the claims, important baselines may be missing, proofs contain important ideas but lack rigor, algorithmic details are only discussed superficially, references are imprecise, assumptions are not sufficiently motivated or explicated, etc.).

**Q2-4 Reproducibility:**

2: Fair: key resources (e.g. proofs, code, data) are unavailable but key details (e.g. proof sketches, experimental setup) are sufficiently well-described for an expert to confidently reproduce the main results.

**Q3 Main Strengths:**

- The rationale behind the construction of such a loss function is well explained.
- The simulation study seems to be robust.

**Q4 Main Weakness:**

- The proposed approach is missing any theoretical guarantee. The only proposition in the paper doesn’t really help to understand in which situations one could expect this approach to be useful and in which not.
- Some of the claims are not backed by appropriate evidence. Like the one on the interpretability of the groups.

**Q5 Detailed Comments To The Authors:**

Here a few typos and comments.

- models on —> models to
- Fix ref style e.g., ’’Conformal prediction methods Vovk et al. [2005] ‘’—> ‘’Conformal prediction methods [Vovk et al. 2005]’’
- ‘’scores ( P(S|X)) is known ‘’—> ‘’scores (P(S|X)) is known‘’
- In the conclusions, “withing”—> “within”
- “Section 4 leverages pinball loss” —> “THE pinball loss”
- It seems that the proposed approach always uses a smaller number of groups. Is it good or bad? In the limit with 1 group you are only considering marginal cover.
- Isn’t double dipping a problem for this type of approach?
- “the most efficient prediction interval for a given X and mis-coverage level is”… What do you mean by most efficient?
- Is it even possible to learn $P(X, S)$ from data? If so, wouldn’t it be possible to get also (asymptotic) conditional coverage according to (3)?

**Q9 Complying With Reviewing Instructions:**

Yes

---

> ### Author Rebuttal · Authors · 2024-04-07
>
> We thank the reviewer for the time and providing constructive comments. Below we address the reviewer's concerns.
>
> **On the theoretical guarantees.** Proposition 5.1 states that, absent any generalization issues, Algorithm 1 would pick the best predictor of the conditional quantile  $F^{-1}_{S|X}(1-\alpha)$ within its hypothesis class. Although this particular result hinges on the `infinite sample' assumption, we stress that Algorithm 1 also performs group-conditional conformal predictions on each of the recovered groups, which do have finite sample guarantees as requested by the reviewer (Eq. 12).
>
> **On the interpretability of groups.** We consider that the groups are interpretable since these are defined by decision trees with relatively low complexity that directly operate on (interpretable input features. Figure 3 in the appendix shows the learned groups in terms of Boolean rules on the input features. For our tabular experiments, the base features are interpretable, and the tree model provides clear decision rules to identify each group. Moreover, for each leaf/group we know its size (\% of samples) and the mean width (efficiency) of the conformal prediction intervals. This information can be used by a data collection or feature conditional generation process that aims to obtain more samples from the unprivileged groups to try to improve their corresponding conformal prediction intervals efficiency and non-conformity score quantile estimates.
>
> **On the typos.** We will address all the typos pointed by the reviewer.
>
> **On the group size of the proposed solutions.** The proposed approach tends to be more conservative in terms of number of groups that competing approaches. Moreover, these groups tend to be meaningful in the sense that all of the identified groups improve their under-coverage w.r.t. a single threshold SCP (1 group). This is captured by MCR score. Moreover, we observe the group with worst under-coverage is considerably closer to the target coverage $0.9$.
>
> **On double dipping.** We are unsure on what is meant by this question. If this pertains to the reuse of the calibration data to both learn the decision tree and later conformalize the prediction, we will add a detailed explanation on why this is not the case. Essentially, the initial tree learns both a partition (the groups) and a prediction that is later thrown out, however, given any grouping function, the (calibration) data is still exchangeable with a new test sample.
>
>
> **Efficiency clarification.** Efficiency refers to the size of the conformal set/interval.
>
> **On the possibility of learning $P(X,S)$ from data and providing prediction intervals with asymptotic conditional coverage guarantees.** Yes, for instance the method proposed by [Amoukou and Brunel, 2023] (included in our comparisons) learns an input-dependent adaptive correction for the calibration non-conformity scores (by approximating $F(S|X)$) that produces prediction intervals with asymptotic conditional coverage guarantees. Our proposed approach has group conditional finite sample guarantees.

---

### Official Review · Reviewer_3x8M · 2024-03-10

**Q2-1 Originality-Novelty:** 2
**Q2-2 Correctness-Technical Quality:** 1
**Q2-5 Clarity Of Writing:** 2

**Q10 Ethical Concerns:**

No ethical concerns.

**Q1 Summary And Contributions:**

The paper aims to extend the marginal guarantees of conformal prediction methods. More precisely, it presents a novel approach that learns interpretable partition functions of the feature space. This allows identifying areas of the input space where the non-conformity scores are as homogeneous as possible when conditioning to the group.

**Q2-3 Extent To Which Claims Are Supported By Evidence:**

1: Poor: the authors fail to convincingly backup their main claims (e.g., if the experimental evaluation is flawed, proofs are lacking or invalid, references are missing, assumptions are not realistic, not specified, or not motivated).

**Q2-4 Reproducibility:**

1: Poor: key details (e.g. proof sketches, experimental setup) are incomplete/unclear, or key resources (e.g. proofs, code, data) are unavailable.

**Q3 Main Strengths:**

The main strengths are:
1. Relevance of the topic:  the paper aims to tackle an important issue of conformal prediction, i.e. the fact that guarantees hold at marginal level. Hence, I think the work is well motivated.
2. Focus on Interpretable partitioning function: the idea of considering groups that are by default interpretable is interesting, as it might provide practical insights to practitioners.
3. Related Work: the work is well positioned within the current literature.

**Q4 Main Weakness:**

These are the main weaknesses of the work:
1. Assumptions behind the main theoretical result: See detailed comments
2. Limited Empirical Evaluation: See detailed comments

**Q5 Detailed Comments To The Authors:**

I have a few concerns about the technical aspects of the work, which I will try to detail here.

Regarding W1:
First, the theoretical guarantees of Proposition 5.1. rely on the infinite sample regime, which is a really strong assumption. Hence, I expected some critical remarks on this assumption by the authors.
Moreover, to the best of my knowledge, conformal prediction’s main goal is to provide valid guarantees in finite samples. More precisely, it seems to me that your goal is to define areas of the input space where the coverage guarantees hold at group level. Hence, I wonder what is the point of adding some theoretical results that require an infinite sample regime. In my opinion, the authors should clearly frame their contribution as a heuristics without providing theoretical results relying on assumptions that are never met (or at least clarify this point). I think this is a discrepancy that undermines the whole work, and the authors should take this into account.

Regarding W2:
I think that to guarantee reproducibility of your results, more details should be provided. First, I think that the hyperparameters choices should be discussed further. For instance, what parameters did you optimize using Optuna? The lack of practical details hinders the overall reproducibility. This is especially relevant as the code is not provided.

Regarding Table 1, why did you choose to highlight in bold the closest value to “the smallest average coverage above the objective (0.9)”. Why did you not highlight the one with the highest average coverage?

**Q9 Complying With Reviewing Instructions:**

Yes

---

> ### Author Rebuttal · Authors · 2024-04-07
>
> We thank the reviewer for the time and constructive comments. Below we address the reviewer's concerns.
>
> **On theoretical results (W1).** Proposition 5.1. states that, absent any generalization issues, Algorithm 1 would pick the best predictor of the conditional quantile  $F^{-1}_{S|X}(1-\alpha)$ within its hypothesis class. We consider this to be a necessary condition for the proposed approach (showing the algorithm would produce the least regularized, best performing model in the asymptotic regime). Although this particular result hinges on the `infinite sample' assumption, we stress that Algorithm 1 also performs group-conditional conformal predictions on each of the recovered groups (last step in Algorithm 1), which do have finite sample guarantees as requested by the reviewer (see Eq. 12). We will stress this in the updated manuscript.
>
> **On empirical evaluation details (W2).** All code necessary to replicate these results will be made available in an open source library after acceptance. Additionally, we will add the specific Optuna configuration for the LGBM surrogate model ($h$ in Eq.13) in the Appendix. To learn the decision tree approximating $h$, we set a minimum of 50 samples per leaf, maximum depth of 5 (we consider that deeper trees are harder to interpret), and set the cost complexity pruning variable as the regularization parameter $\theta$ with $\theta_0 = 1e-5$ and $\Delta_{\theta_{t}} = 9\times\theta_{t}$.
>
> **Regarding Table 1.** We chose to highlight, from within the models achieving or exceeding the target coverage, the one with the smallest coverage since this model would also have the highest efficiency (smaller prediction sets) while still satisfying coverage objective $>0.9$. For instance, an infinite width predictive interval would maximize target coverage but have poor efficiency/utility. We will expand the table in the appendix to include the mean prediction interval width as a reference to highlight this observation.

---

### Official Review · Reviewer_Yovg · 2024-03-20

**Q2-1 Originality-Novelty:** 3
**Q2-2 Correctness-Technical Quality:** 3
**Q2-5 Clarity Of Writing:** 2

**Q10 Ethical Concerns:**

No. I do not see any potential ethical or societal issues related to this work.

**Q1 Summary And Contributions:**

The paper proposes an optimization strategy to partition the input space for localizing Conformal Prediction. The space partition is built to make the value of the conformity scores homogeneous within single groups. In practice, the problem is cast as a bilevel optimization over the complexity of a decision tree trained with the pinball loss.

**Q2-3 Extent To Which Claims Are Supported By Evidence:**

3: Good: the main claims are supported by convincing evidence (in the form of adequate experimental evaluation, proofs, (pseudo-)code, references, assumptions).

**Q2-4 Reproducibility:**

3: Good: key resources (e.g. proofs, code, data) are available and key details (e.g. proofs, experimental setup) are sufficiently well-described for competent researchers to confidently reproduce the main results.

**Q3 Main Strengths:**

- Group-CP is a scalable approximation of the unachievable conditional CP. Improving grouping techniques may have a great impact on
real-world application with large feature space.
- Enforcing the conformity scores to be similar within the same group is a nice idea. Measuring the distance between empirical coverage and the approximated quantile is also ingenious.
- The adversarial interpretation of the optimization scheme is interesting.

**Q4 Main Weakness:**

- The language may be improved. The paper contains ambiguous sentences, e.g. *Since these learned groups are expressed as strictly a function of the input, they can be used for downstream tasks such as data collection or model selection.*'
- The optimization problem does not look easy.
- The proposed method is an approximation of an ideal input-conditional CP algorithm. The authors do not show, either theoretically or empirically, if the optimized models outperform other existing approximation schemes.

**Q5 Detailed Comments To The Authors:**

- Is this the first time adversarial optimization is used to boost the efficiency of CP?
- How does minimizing group mis-coverage ensure the conformity scores are homogenous inside a group?
- Does ${\cal Y}$ need to be continuous?
- How expansive is the optimization?
- Is $\theta$ a 1-dimensional parameter? Is this the only free parameter of the outer optimization?
- MRC does not depend explicitly on theta. Is it minimal for the most or least complex tree?

**Q9 Complying With Reviewing Instructions:**

Yes

---

> ### Author Rebuttal · Authors · 2024-04-07
>
> We thank the reviewer for the time dedicated to read the paper and providing constructive comments. Below we address the reviewer's concerns.
>
> **On the comment `since these learned groups are expressed as strictly a function of the input...'.**  Figure 3 in the appendix shows the discovered groups in terms of Boolean rules on the input features. For our tabular experiments, the base features are interpretable, and the tree model provides clear decision rules to identify each group. Moreover, for each leaf/group we know its size (\% of samples) and the mean width (efficiency) of the conformal prediction intervals. This information can be used by a data collection or feature conditional generation process that aims to obtain more samples from the unprivileged groups to try to improve their corresponding conformal prediction intervals efficiency and non-conformity score quantile estimates. For example, in the Concrete dataset (Figure 3.b) we have a small group (11\%) with the largest prediction interval mean width where the cement in mixture is above 426 kg/m3. A reasonable strategy for a data collection approach would be to gather more samples with these characteristics to see if the efficiency and/or miscoverage can be improved.
>
> **On optimization hardness.** In this work we used simple line-search as our search algorithm since the model's regularization were well characterized by a single parameter. For more complex settings, one could leverage existing hyperparameter optimization frameworks to directly optimize the proposed MCR score.
>
> **On the comparison with other input-conditional CP algorithms.** In our experiments we compare against two of the approaches proposed by Amoukou and Brunel [2023], LCP-RF-G, RF-G, these are input-conditional split CP algorithms that use quantile random forest (QRF) to learn a weighting function of the non-conformity scores that takes the covariates X as inputs. Then they use a clustering approach to detect the communities of the QRF's weights. It is important to note that the QRF algorithm [Meinshausen, N. and Ridgeway 2006.] does not minimize (an approximation of) the quantile objective ($1-\alpha$) but instead is minimizes the inter-leaf variance of the non-conformity scores; the leaves of this QRF algorithm store the entire list of non-conformity scores of train samples falling in the leave, rather than a single summary statistic. In our formulation the partition functions that we propose are approximations of the $1-\alpha$ quantile of the non-conformity score since they are obtained minimizing the pinball loss.
>
> Meinshausen, N. and Ridgeway, G. Quantile regression forests. Journal of Machine Learning Research, 7(6), 2006.
>
> **On works for adversarial optimization.** While there are works such as [Bastani et all 2022] that propose adversarial optimization techniques to tackle distribution shift in the context of sequential conformal prediction. To the best of our knowledge there is no method that considers the adversarial setting for proposing group partitions in the context of conformal prediction.
>
> **On homogeneity of miscoverage within a group.** The homogeneity of the quantile of the non-conformity scores inside a group is achieved because each proposed decision tree minimizes pinball loss, and the predictions are leaf-wise constant. The group mis-coverage ratio is a conservative metric to control for generalization problems, and is used to select amongst each decision tree; it measures the ratio between worst group miscoverage of the current model to the one achieved by an interpretable baseline.
>
> **On the continuity of Y.** Y does not need to be continuous, the same approach can be applied to, for example, a classification specific non-conformity score such as those proposed by [Sadinle et al., 2019 ; Romano et al 2020].
>
> Yaniv Romano, Matteo Sesia, and Emmanuel J. Candès. Classiﬁcation with valid and adaptive coverage. Advances in Neural Information Processing Systems (NeurIPS), 2020.
> Mauricio Sadinle, Jing Lei, and Larry Wasserman. Least ambiguous set-valued classiﬁers with bounded error levels. Journal ofthe American Statistical Association, 2019.
>
> **On optimization expense.** None of our experiments on the evaluated datasets took more than 10 minutes to run. We will add compute time for all datasets in the appendix.
>
> **On MCR dependence on theta.** MCR implicitly depends on $\theta$ through $\tau_{\theta}$ which is the tree that minimizes the pinball loss with regularization parameter $\theta$. It is not necessarily minimal for the least complex tree, for instance if a tree consist of a single leaf/group the MCR = 1 which is equivalent to the standard SCP. Moreover, for more complex trees it is highly likely (and observed in practice) that at least one group fails to generalize to unseen data and therefore attains an MCR >>1. In our experiments we observe that the curve of the MCR is `bowl shaped' with $\theta$.

---

### Official Review · Reviewer_6p43 · 2024-03-23

**Q2-1 Originality-Novelty:** 3
**Q2-2 Correctness-Technical Quality:** 3
**Q2-5 Clarity Of Writing:** 3

**Q10 Ethical Concerns:**

No.

**Q1 Summary And Contributions:**

The paper proposes a new method for coming up with partitions of the domain, for obtaining conformal coverage guarantees with better efficiency and that would hold conditionally on various interpretable regions of the domain. The partitions have the flavor of clustering, having the aim to make sure that (1) the quantiles of nonconformity scores on each region are as tight-knit/similar as possible, and (2) different regions have different associated quantiles. They explore the implementation's coverage, efficiency and interpretability performance compared to other methods (including a similar recent one), and observe favorable performance.

**Q2-3 Extent To Which Claims Are Supported By Evidence:**

3: Good: the main claims are supported by convincing evidence (in the form of adequate experimental evaluation, proofs, (pseudo-)code, references, assumptions).

**Q2-4 Reproducibility:**

3: Good: key resources (e.g. proofs, code, data) are available and key details (e.g. proofs, experimental setup) are sufficiently well-described for competent researchers to confidently reproduce the main results.

**Q3 Main Strengths:**

--- The authors' approach of coming up with what essentially amounts to data driven binning of the domain is quite novel on the technical side, involving an interesting metric of goodness for partitions (the group miscoverage ratio MCR).

--- Furthermore, the method appears to be effective in experiments: it produces prediction sets with adaptive coverage when used in conjunction with applying the simplest possible split conformal method on top of its generated partitions.

--- The claim of the regions being interpretable also seems to hold in the proposed experiments.

**Q4 Main Weakness:**

There are no big weaknesses. As I discuss below, there are:

--- Somewhat deeper discussion of related work to be had (in particular, a discussion of the technical and empirical differences to the benchmark [Amoukou and Brunel, 2023])
--- Possible extensions to the experimental setup to verify the empirical versatility and promise of the proposed approach (Exploring different models/datasets/settings to investigate the interpretability/goodness of groups),
--- Minor presentation points to be improved upon,

But overall I am (nearly) satisfied with the manuscript as it is.

**Q5 Detailed Comments To The Authors:**

--- First off, I would appreciate some more experimental insights into the method. Note, for instance, that in the current setup you utilize a distance function that penalizes undercoverage but not overcoverage. By including overcoverage as well, can you get tighter prediction sets?

--- Also, the current experimental outcomes, including the interpretability-confirming diagram in the Appendix, seem to rest to a significant extent on the power of the underlying model, which in this instance is a GBT. Would a less powerful model with a less tight (but still decent) fit produce worse, more noisy and less interpretable sets? Or would it lead to simpler, more interpretable but slightly less performant (coverage and efficiency-wise) partitions? That sort of an ablation study would really help drive home the interpretability benefits of the proposed method.

--- It is also stated at the end of Section 6 that "groups with higher uncertainty (larger mean width) tend to have a smaller size". This seems like a nontrivial observation to explain; what might be at play is how "easy" the dataset is vs. how powerful the underlying model is, or some heterogeneity in the domain. Or, possibly, it might be related to the proposed algorithm being good at breaking up large uncertainty-riddled regions in the tentative partition down into more manageable chunks. Is it possible to dig into this a bit further?

--- Some relevant pieces of literature should I think be discussed in more detail:

(a) First of all, the main baseline/closest method to the proposed one here, namely [Amoukou and Brunel, 2023], should be discussed in more than just the few brief sentences in Section 3. I would like to see a more detailed comparison of the techniques, as well as (optionally) speculation on why the current method might be overperforming the previous one.

(b) Secondly, the objective of finding constant-quantile partition regions, each ultimately having the same target coverage, seems intimately related to (or possibly the same as) the notion of threshold-conditional coverage introduced in [Bastani et al, 2022, Jung et al, 2023]: for a given binning of the space of nonconformity score quantiles, it requires the coverage to be at the target for every binned threshold value. In this context, it would seem that the present paper develops a method for data-driven calibration binning. If that is the case, then I would like this to be discussed further as well, to further contextualize the propose method.

--- The presentation is good overall, but there are aspects that could improve it somewhat. For instance, Figure 1b and 1c could be enlarged and displayed a bit more prominently/even earlier on, as I think they provide the maximum amount of intuition about the method's objective. Figure 1a, despite the effort in spelling out the different components of the setup, may look a bit more mysterious to those less familiar with group-conditional coverage. Another comment: it is stated several times early on that the proposed partitioning approach can compose nicely with existing group conditional approaches. However, as described in the paper, the current pipeline for producing and putting into action the partition into groups interacts most naturally with just running separate split conformal procedures on each element of the partition (which is in fact what is done in the experiments), so I think this can be clarified as soon as possible in the paper.

**Q9 Complying With Reviewing Instructions:**

Yes

---

> ### Author Rebuttal · Authors · 2024-04-07
>
> We thank the reviewer for the time and constructive and insightful comments.  Below we address the reviewer's questions.
>
>
> **On the choice of distance function**. We opted for
> $d(1-\alpha,p) = (1-\alpha - p)_{+}  $
>
> because it aligned with our conservative approach by minimizing violation in terms of undercoverage. We will add additional experimental results for the use of $d(1-\alpha,p) = |1-\alpha - p|$ in the MCR score, which equally penalizes over- and under-coverage gaps. This could lead to a model with better efficiency (tighter prediction sets), better absolute miscoverage overall, but potentially worse undercoverage. We want to highlight that the models that we rank ($\tau_{\theta}$) minimize the pinball loss for the $1-\alpha$-th quantile, and therefore approximate the conditional quantile function $F_{S|X}^{-1}(1-\alpha)$, which is the optimal solution in terms of efficiency and coverage.
>
>
>
> **On the choice of base model and interpretability of sets.** Table 2 in the Appendix shows performance of the proposed approach where we learn conformal prediction intervals for a LASSO regression model. We observe that the number of discovered groups is larger than those of a GBT regression model for the same dataset (Table 1). In most cases, the GBT model is equal or better than LASSO in terms of r2 score, and therefore reduces the unexplained variance of the target $Y|X$. This leads to less regions of different uncertainty and tighter prediction sets. We will include a plot similar to Figure 2 and 4 where we show the joint distribution of interval width and coverage for LASSO and GBT model.
>
>
> **On the width of high uncertainty groups.** We examined our results and observe that the base GBM model fails to generalize (gap between train and test performance was larger) in the high uncertainty regions. The LASSO models do not suffer from this, the discovered regions have worse performance overall without significant train-test disparities. We will include in the Appendix this analysis and corresponding discussion.
>
> **On comparisons  with [Amoukou and Brunel, 2023].** We will add a more detailed description of the method in the main text. In summary, the method proposed by [Amoukou and Brunel, 2023] use a quantile random forest (QRF) that approximates the distribution of the non-conformity scores on the calibration dataset. Then they use a clustering approach to detect the communities of the QRF's weights. It is important to note that the QRF algorithm [Meinshausen, N. and Ridgeway 2006.] does not minimize (an approximation of) the quantile objective ($1-\alpha$) but instead is minimizes the inter-leaf variance of the non-conformity scores; the leaves of this QRF algorithm store the entire list of non-conformity scores of train samples falling in the leave, rather than a single summary statistic. In our formulation the partition functions that we propose are approximations of the $1-\alpha$ quantile of the non-conformity score since they are obtained minimizing the pinball loss.
>
> Meinshausen, N. and Ridgeway, G. Quantile regression forests. Journal of Machine Learning Research, 7(6), 2006.
>
> **On relations with threshold-conditional coverage.** We agree with the observation that our method satisfies threshold-conditional coverage. We will add this interpretation to the manuscript!
>
> **On the presentation improvements.** We will make the suggested changes in Figure 1 and clarify that the proposed group discovery approach interacts most naturally with group conditional scp.

---

### Meta-Review · Area_Chair_dVg4 · 2024-04-16

The paper presents interesting idea to obtain partitions and group-wise coverage in an optimal way. The reviewers all agree that the paper presents an interesting problem, as well as some solutions that appear empirically convincing.

There is still some unclarity as to how the theoretical proposition is related to the particular algorithm proposed, as well as the specific model considered, resulting in a methodological solution that appears satisfactory yet heuristic. The answers of the reviewers convinced the reviewers, at least to increase their score and reach a consensus towards acceptance (even if sometimes weak).